# Which breathlessness dimensions associate most strongly with fatigue?–The population-based VASCOL study of elderly men

**Lucas Cristea**[1,2][◉]*, **Max Olsson**[1][◉], **Jacob Sandberg**[1][◉], **Slavica Kochovska**[3][◉], **David Currow**[3][◉], **Magnus Ekström**[1][◉]

**1** Respiratory Medicine, Allergology and Palliative Medicine, Department of Clinical Sciences Lund, Lund University, Lund, Sweden, **2** Kallinge Health Center, Kallinge, Sweden, **3** Faculty of Science, Medicine and Health, University of Wollongong, Wollongong, New South Wales, Australia

◉ These authors contributed equally to this work.
* lucas.cristea@med.lu.se

## Abstract

### Background

Breathlessness and fatigue are common symptoms in older people. We aimed to evaluate how different breathlessness dimensions (overall intensity, unpleasantness, sensory descriptors, emotional responses) were associated with fatigue in elderly men.

### Methods

This was a cross-sectional analysis of the population-based VAScular disease and Chronic Obstructive Lung Disease (VASCOL) study of 73-year old men. Breathlessness dimensions were assessed using the Dyspnoea-12 (D-12), Multidimensional Dyspnoea Profile (MDP), and the modified Medical Research Council (mMRC) scale. Fatigue was assessed using the Functional Assessment of Chronic Illness Therapy–Fatigue (FACIT-F) questionnaire. Clinically relevant fatigue was defined as FACIT-F ≤ 30 units. Scores were compared standardized as z-scores and analysed using linear regression, adjusted for body mass index, smoking, depression, cancer, sleep apnoea, prior cardiac surgery, respiratory and cardiovascular disease.

### Results

Of 677 participants, 11.7% had clinically relevant fatigue. Higher breathlessness scores were associated with having worse fatigue; for D-12 total, -0.35 ([95% CI] -0.41 to -0.30) and for MDP A1, -0.24 (-0.30 to -0.18). Associations were similar across all the evaluated breathlessness dimensions even when adjusting for the potential confounders.

### Conclusion

Breathlessness assessed using D-12 and MDP was associated with worse fatigue in elderly men, similarly across different breathlessness dimensions.

⬚ OPEN ACCESS

**Data Availability Statement:** As stated by Lund University's ethical body approving the data collection (Dnr 2008/676), and the Swedish Ethical

Review Authority analysis (Dnr 2019-00134) approving the analysis of this study the data is not allowed to be shared publicly. The participants were also promised when agreeing to participate in the VASCOL study that their data should not be shared publicly. Sharing sensitive data such as health data publicly is not complying to article 9 of the General Data Protection Regulation (EU 2016/679) as this would compromise the privacy of the participants. The General Data Protection Regulation (EU 2016/679) also considers de-identified sensitive data as sufficient to risk the privacy of participants. According to Swedish law (2003:460) concerning research including humans, ethical permission is required to process data including humans. To access the data from the VASCOL study, ethical approval first needs to be required from the Swedish Ethical Review Authority (https://etikprovningsmyndigheten.se). Researchers can then contact the principal investigator (Magnus Ekström) with suggestions for analysis - these suggestions will then be discussed in the VASCOL research group.

**Funding:** The study was funded by unrestricted grants from the Research Council of the Region of Blekinge, and from the Swedish Research Council (Dnr: 2019-02081). The funders had no role in study design, data collection and analysis, decision to publish, or preparation of the manuscript.

**Competing interests:** Dr. Currow is an unpaid advisory board member for Helsinn Pharmaceuticals. He is a paid consultant and receives payment for intellectual property with Mayne Pharma and a consultant with Specialised Therapeutics Australia Pty. Ltd. The remaining authors report no competing interests in any way. This does not alter our adherence to PLOS ONE policies on sharing data and materials.

# Introduction

Chronic breathlessness and fatigue are main symptoms in several severe illnesses such as cardiorespiratory disease and cancer. Chronic breathlessness is experienced by 9–11% of adults in the community [1, 2], with prevalence rising among the elderly (>70 years of age) where it affects 25–32% in their daily life [3, 4]. The symptom is often not recognized by health care professionals and is, in that regard, an invisible and neglected symptom [5–7]. The symptoms are associated with ongoing suffering, reduced function [8], poorer prognosis [9], anxiety and depression [10], fear [11] and worse quality of life (QoL) [12]. Breathlessness is a multidimensional symptom that is influenced by cognitive, psychological, physiological, and environmental factors [11, 13]. To capture the relevant aspects of the symptom, multiple dimensions may need to be considered in its assessment. Different dimensions of breathlessness can be measured with the Dyspnoea-12 (D-12) [14, 15] and Multidimensional Dyspnoea Profile (MDP) questionnaires [16].

Fatigue is a sensation described as an unusual and overwhelming tiredness that cannot be alleviated by sleep [17] and is not the same as physiological exhaustion after physical or mental effort. It is a common symptom in severe diseases and is reported to be associated with breathlessness, such as in people with chronic obstructive pulmonary disease (COPD) and idiopathic pulmonary fibrosis (ILD) [18, 19]. Fatigue can be measured by the validated Functional assessment of chronic illness–fatigue (FACIT-F) questionnaire [20].

No study to date has evaluated if some dimensions of breathlessness are more strongly associated with fatigue than others, and some symptom dimensions may be more strongly linked than others to worse fatigue. This knowledge is important as it might affect clinicians' assessments of patients and the treatments offered subsequently. We hypothesize that there is a difference in how D-12 and MDP associate with fatigue.

The aim of the study was to evaluate how different dimensions of breathlessness were associated with fatigue in elderly, community-dwelling men.

# Methods

## Study design and population

This was a cross-sectional, population-based analysis of the VAScular disease and Chronic Obstructive Lung disease (VASCOL) study of 73-year-old men. VASCOL included 1,302 men aged 65 years in 2011–2012 who participated in screening for aortic aneurysm and who consented to participate in a longitudinal follow-up study. The design and measurements of the VASCOL study have been detailed elsewhere [21]. In 2010–2011, 1900 men aged 65 years in Blekinge, Sweden were invited to a screening campaign of abdominal aortic aneurysm–those men were invited to participate in the VASCOL study (ongoing longitudinal epidemiological cohort study). VASCOL was based on physiological measurements, but also completion of self-reported surveys–including information about multidimensional breathlessness and fatigue (through D-12, MDP and FACIT-F). Data was collected in 2010–2011, and the same men (who were still alive and with a known address) were asked in 2019 to respond to a follow-up postal survey.

No patient-reported outcomes pertaining to breathlessness were assessed at baseline. The present study analysed data from this 2019 follow-up. The analysis included participants (677 men) with completed data on the breathlessness (D-12 and MDP) and fatigue (FACIT-F) measurements, and who were recruited between 2019-03-01 to 2019-09-28. The study is reported in accordance with the Strengthening the Reporting of Observational studies in Epidemiology (STROBE) guidelines [22].

## Assessments

Descriptive data (categorized as continuous and categorical) including self-reported height (cm), weight (kg), smoking status (current, former or never-smoker), smoking exposure (years of smoking and average number of cigarettes per day), and the presence of physician-diagnosed conditions were dichotomised (asthma, COPD, angina pectoris, atrial fibrillation, heart failure, myocardial infarction, valvopathies, diabetes mellitus, rheumatologic disease and stroke) or none/other [21].

**Breathlessness.** Breathlessness was measured using the D-12 and MDP questionnaires, as well as the modified Medical Research Council (mMRC) breathlessness scales [23], over a 14-day recall period.

The D-12 questionnaire comprises 12 items (descriptors), each scored on a 4-point scale of 0 (none), 1 (mild), 2 (moderate), or 3 (severe) (10). The first seven items pertain to the physical domain (D-12 physical) of breathlessness, while the remaining five items pertain to the affective domain (D-12 affective). The range for D-12 total score is 0–36, with 0–21 for the physical score and 0–15 for the affective score. Higher scores indicate worse breathlessness [14]. D-12 has been validated to be completed as a postal questionnaire in population-based studies [23, 24].

The MDP comprises 11 items rated on 0–10 numerical rating scales (NRS) and evaluated across three domains: 1) the MDP A1 which is the total unpleasantness or discomfort of breathing (range 0–10); 2) the MDP immediate perception score (range 0–60) which is the sum of A1 and the intensities of five sensory qualities (muscle work or effort; air hunger; chest tightness or constriction; mental effort or concentration; and breathing a lot); and 3) the MDP emotional response score (range 0–50) which is the sum of the intensities for each of five emotional responses (depression; anxiety; frustration; anger; and fright). Higher scores reflect worse breathlessness [25]. MDP has also been validated to be completed as a postal questionnaire in population-based studies [16].

The mMRC breathlessness scale is an ordinal questionnaire which measures the level of exertion that generates breathlessness. The scale ranges from 0 to 4, where higher grade correlates to less exertion before breathlessness supervenes [26]. mMRC 1 is defined as breathlessness when hurrying or walking up a slight hill, mMRC 2 pertains to walking slower than people of the same age due to breathlessness or having to stop to breath when walking at own pace, mMRC 3 consists of stopping for breath after walking about 100 yards or after a few minutes on level ground and mMRC 4 is defined as being too breathless to leave the house or being breathless when dressing [27].

**Fatigue.** Fatigue was measured using the FACIT-F questionnaire, which is a validated self-reported scale based on a 7-day recall period [28]. FACIT-F comprises 13 items each scored on a 5-point scale of 4 (not at all), 3 (a little bit), 2 (some-what), 1 (quite a bit) and 0 (very much). The questions relate to physical, social, emotional, and functional well-being in relation to illness [18]. The FACIT-F total score ranges 0–52, with lower scores reflecting more severe fatigue, and values $\leq 30$ indicating fatigue that is likely to be clinically significant [29].

## Statistical analyses

Characteristics of the participants were tabulated in 2019 (answering the postal questionnaire). No data were imputed. Stata version 14 was used for analysis.

Associations with fatigue (FACIT-F) were analysed for each breathlessness score of MDP (overall unpleasantness [A1], perception, and emotional response scores) and D-12 (total, physical, and affective scores) using linear regression. When looking at association of breathlessness with clinically relevant fatigue, logistical regression was used. Analyses were performed unadjusted and adjusted for confounders, which were selected based on subject matter

knowledge and previous studies [30, 31]–body mass index (BMI), smoking status, pack-years of smoking, presence of physician-diagnosed cardiovascular disease (any of myocardial infarction, heart failure, valvopathies, atrial fibrillation, or stroke), respiratory disease (COPD, asthma, or other lung disease), depression [32], cancer [33], sleep apnoea [34], and prior cardiac surgery [35, 36].

Strength of association with fatigue was compared between the breathlessness dimension scores. To make estimates comparable across the different scales, all breathlessness scores were log transformed, to obtain a more normal distribution, and converted into z-scores (calculated as [raw score–mean] / standard deviation [SD] of the score). Using z-scores is an established method to enable comparisons of scores across scales and was used in an analysis of breathlessness dimensions in relation to QoL [12]. The fatigue score was not transformed, and only the breathlessness scores were log- and z-transformed.

## Results

### Participants

Of the 1,302 participants in the initial VASCOL population sample from 2011–2012, 1,193 (92%) participants were still alive and had a known address in 2019. Of these, 907 (76%) participated in the 2019 follow-up by returning the questionnaire. After exclusion of 230 participants due to missing data on D-12, MDP and FACIT-F, a total of 677 participants were included in the analysis.

Participant characteristics are shown in Table 1. All participants were 73-years old men; a breathlessness score was reported by 215 (33%) for mMRC ≥1, 202 (30%) for D-12 total and by 103 (15%) for MDP A1. Moderate to severe breathlessness, defined as MDP A1 ≥ 4, was reported by 29 (4.3%). Clinically relevant fatigue, defined as FACIT-F ≤ 30, was only reported by 79 (11.7%) participants. Of those, 24 (30.4%) participants were more breathless with an mMRC score of 2 and 3 while 18 (22.8%) participants had the most severe breathlessness with mMRC of 4.

### Breathlessness dimensions and fatigue

Associations between different breathlessness dimension scores with fatigue, unadjusted and adjusted for confounders, are shown in Table 2. The associations between D-12 total score and fatigue can be seen in Fig 1, and the associations between MDP A1 and fatigue can be seen in Fig 2. The association with fatigue was similar for D-12 total (-0.35; 95% CI, -0.41 to -0.30) and MDP A1 (-0.24; 95% CI, -0.30 to -0.18). The associations were similar across the different breathlessness dimension scores (Fig 2). Breathlessness was associated with increasing fatigue across all dimension scores, but it was not a strong association even though breathlessness and fatigue often co-exist. The associations between fatigue, D-12 and MDP are also shown in Fig 3. The variance of the fatigue scores is between 54–60% for the models with D-12 and MDP while adjusting for the confounders, and the models were very similar–the remaining percentage is due to factors we did not measure.

Table 3 shows the association for each breathlessness dimension with clinically relevant fatigue. Similarly, to Table 2, we can see that both D-12 and MDP show a similar association with clinically relevant fatigue even after adjusting for confounders D-12 total: OR 2.46 (95% CI, 1.79–3.38) and MDP A1: OR 1.85 (95% CI, 1.34 to 2.57).

### Main findings

Breathlessness was associated with increasing fatigue across all dimension scores in elderly men, but no breathlessness dimension was more strongly associated with fatigue than the others and the associations were only modest.

**Table 1. Characteristics of 677 men aged 73 years from the general population.**

| Factor | With clinically significant fatigue* | Without clinically significant fatigue* | All |
|---|---|---|---|
| | n = 79 | n = 598 | N = 677 |
| Age, mean (SD) | 73.27 (0.74) | 73.22 (0.68) | 73.2 (0.7) |
| BMI, mean (SD) | 29.61 (4.86) | 28.01 (3.92) | 28.2 (4.07) |
| Ever smoked, n (%) | 59 (74.7) | 381 (63.7) | 440 (65.0) |
| Missing | 1 (1.3) | 7 (1.2) | 8 (1.2) |
| Pack-years of smoking, mean (SD) | 7.85 (8.53) | 9.06 (14.54) | 8.92 (14.0) |
| **Morbidities, n (%)** | | | |
| Respiratory disease | 22 (20.3) | 43 (6.9) | 65 (9.6) |
| Asthma | 10 (12.7) | 23 (3.8) | 33 (4.9) |
| COPD | 10 (12.7) | 15 (2.5) | 25 (3.7) |
| Other respiratory diseases | 2 (2.5) | 5 (0.8) | 7 (1.0) |
| Cardiovascular disease | 51 (65.0) | 223 (28.4) | 274 (40.5) |
| Angina pectoris | 10 (12.7) | 37 (6.2) | 47 (6.9) |
| Atrial fibrillation | 19 (24.1) | 84 (14.0) | 103 (15.2) |
| Heart failure | 7 (8.9) | 18 (3.0) | 25 (3.7) |
| Myocardial infarction | 7 (8.9) | 57 (9.5) | 64 (9.5) |
| Valvopathies | 8 (10.1) | 27 (4.5) | 35 (5.2) |
| Diabetes mellitus | 14 (17.7) | 80 (13.4) | 94 (13.9) |
| Rheumatologic disease | 5 (6.3) | 26 (4.3) | 31 (4.6) |
| Stroke | 6 (7.6) | 43 (7.2) | 49 (7.2) |
| **Breathlessness and fatigue** | | | |
| mMRC score, n (%) | | | |
| 0 | 19 (24.1) | 427 (71.4) | 446 (67.5) |
| 1 | 16 (20.3) | 82 (13.7) | 98 (14.8) |
| 2 | 12 (15.2) | 45 (7.5) | 57 (8.6) |
| 3 | 12 (15.2) | 15 (2.5) | 27 (4.1) |
| 4 | 18 (22.8) | 15 (2.5) | 33 (5.0) |
| Missing | 2 (2.5) | 14 (2.3) | 16 (2.4) |
| D-12 total score, mean (SD) | 7.42 (8.21) | 0.95 (2.52) | 1.71 (4.21) |
| D-12 physical score, mean (SD) | 4.46 (4.54) | 0.68 (1.74) | 1.12 (2.55) |
| D-12 affective score, mean (SD) | 2.96 (3.90) | 0.27 (0.96) | 0.59 (1.82) |
| MDP A1 unpleasantness score, mean (SD) | 2.39 (2.24) | 0.49 (1.02) | 0.71 (1.37) |
| MDP perception score, mean (SD) | 10.1 (12.69) | 1.70 (4.35) | 2.68 (6.53) |
| MDP emotional response score, mean (SD) | 7.58 (9.61) | 0.92 (2.83) | 1.70 (4.72) |
| FACIT-F | 22.54 (5.87) | 44.52 (5.48) | 41.96 (8.97) |

*Clinically significant fatigue was defined as having ≤ 30 units on the Functional Assessment of Chronic Illness–Fatigue (FACIT-F) questionnaires [29].

*Abbreviations*: BMI, body mass index; COPD, chronic obstructive pulmonary disease; mMRC, modified Medical Research Council; D-12, Dyspnoae-12; MDP, Multidimensional Dyspnoea Profile; FACIT-F, Functional assessment of chronic illness–fatigue.

## What this study adds

This is the first study to use D-12, MDP and FACIT-F for health assessment of participants in the general population. Our hypothesis that the instruments would differ in reflecting fatigue scores was not supported. Both D-12 and MDP show similar levels of association with a validated measure of fatigue. Both instruments are useful and provide similar information regarding multidimensional breathlessness and fatigue. These findings complement Swigris et al, who showed an association between two breathlessness instruments (the University of

**Table 2. Association for each breathlessness dimension with fatigue.**

| | Association with fatigue (FACIT-F) score (Lower score reflects worse fatigue) | | |
| --- | --- | --- | --- |
| | **Unadjusted** | **Adjusted \*** | **r²-value** |
| | **Beta (95% CI)** | **Beta (95% CI)** | |
| **D-12** | | | |
| Total score | -0.55 (-0.61 to -0.49) | -0.35 (-0.41 to -0.30) | 0.60 |
| Physical score | -0.55 (-0.61 to -0.49) | -0.35 (-0.41 to -0.29) | 0.60 |
| Affective score | -0.50 (-0.56 to -0.44) | -0.31 (-0.37 to -0.26) | 0.59 |
| **MDP** | | | |
| A1 unpleasantness score | -0.48 (-0.55 to -0.42) | -0.24 (-0.30 to -0.18) | 0.54 |
| Perception score | -0.47 (-0.53 to -0.41) | -0.23 (-0.29 to -0.17) | 0.55 |
| Emotional response score | -0.52 (-0.58 to -0.45) | -0.27 (-0.33 to -0.20) | 0.56 |

To be able to compare the strengths of the associations between the different scales, the breathlessness scores were log transformed (to yield more normal distributions) and analysed as z-scores. Associations are by linear regression for each breathlessness score separately, adjusted for confounders. A change in FACIT-F gives one SD change in D-12 and MDP. Lower FACIT-F reflects worse fatigue as seen by a negative association. The r²-value is the variance in fatigue scores between D-12 and MDP models, adjusted for confounders.

\* Adjusted for the confounders: smoking history, packet-years, BMI, respiratory diseases, cardiovascular diseases, depression, cancer, sleep apnoea and cardiac surgery.

*Abbreviations*: D-12, Dyspnoea-12; MDP, Multidimensional Dyspnoea Profile; FACIT-F, Functional Assessment of Chronic Illness–Fatigue. CI, confidence interval. BMI, body mass index.

California San Diego Shortness of Breath Questionnaire (UCSD) and D-12) and fatigue (measured with the Multi-Dimensional Health Assessment Questionnaire (MDHAQ)) in people with connective tissue disease-related interstitial lung disease (CTD-ILD) [37]. Our study

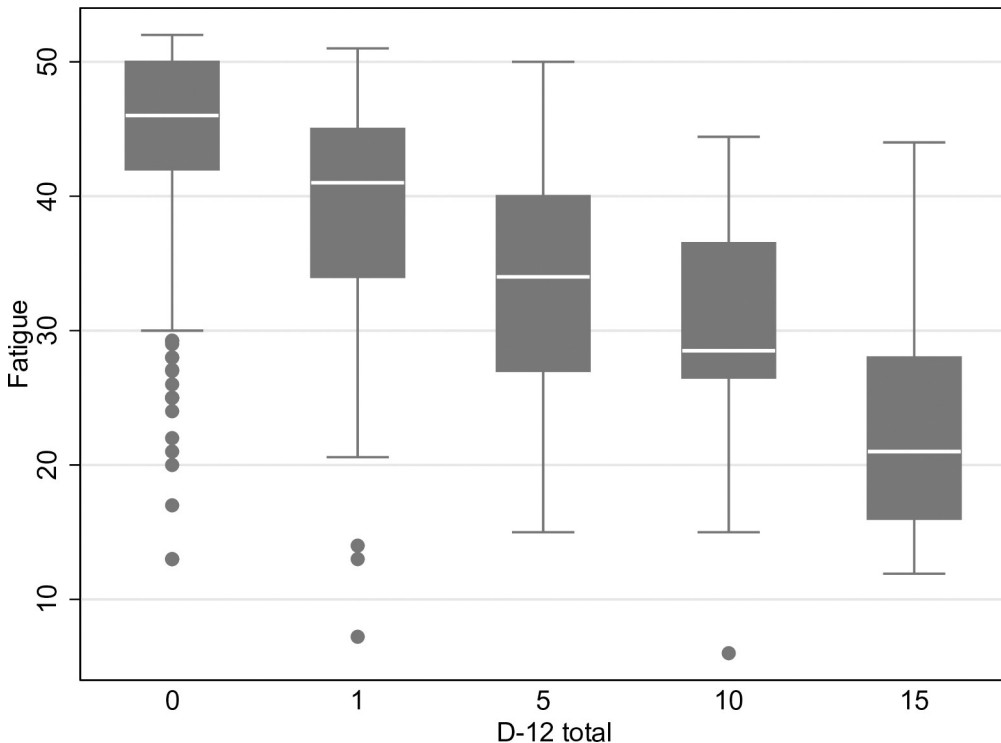

**Fig 1. Fatigue in relation to breathlessness measured using the Dyspnoea-12 (D-12) total score.**

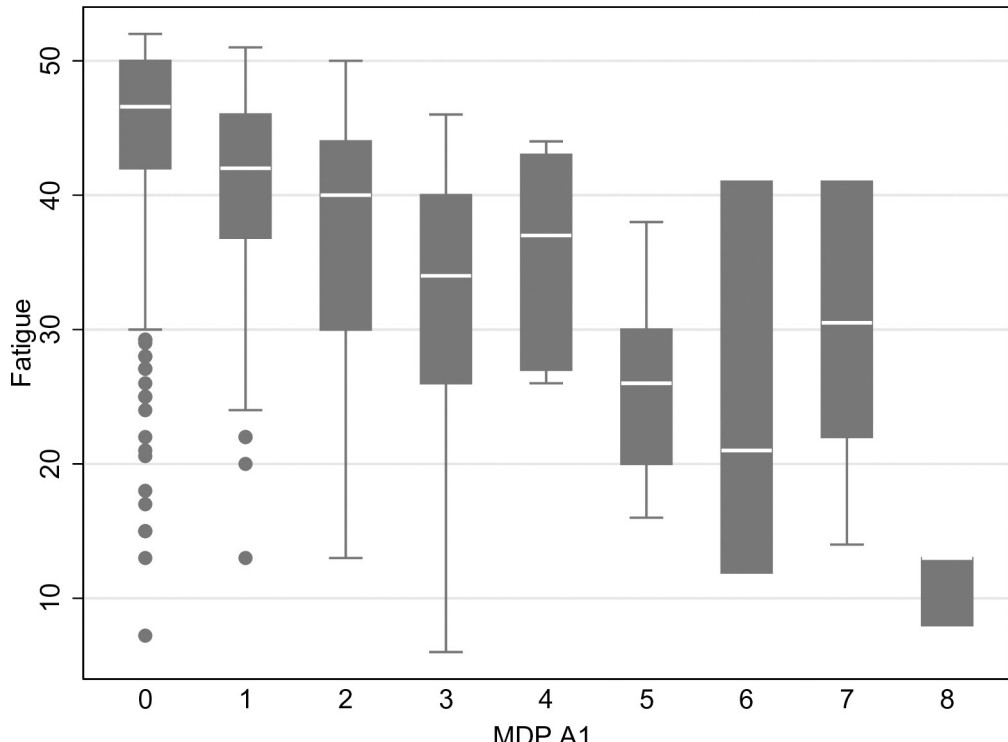

**Fig 2. Fatigue in relation to breathlessness using the Multidimensional Dyspnoea Profile (MDP) overall unpleasantness score.**

added to their findings by evaluating more dimensions and by performing a population-based analysis. Our findings align with those of Pavli et al who showed an association between fatigue and breathlessness in post-COVID syndrome [38].

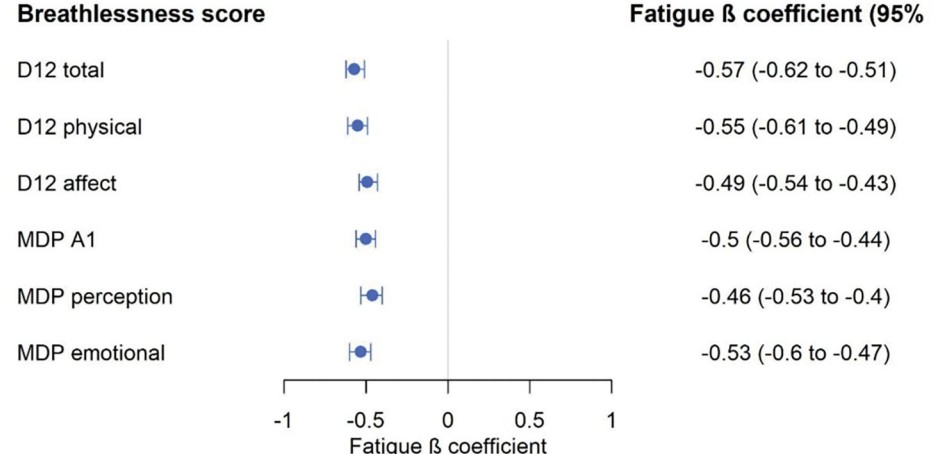

**Fig 3. Forest plot over associations between fatigue, D-12 and MDP.** To be able to compare the strengths of the associations between the different scales, the breathlessness scores were transformed and analysed as z-scores. Associations are by linear regression for each breathlessness score separately, adjusted for confounders (smoking history, packet-years, BMI, respiratory diseases, cardiovascular diseases, depression, cancer, sleep apnoea and cardiac surgery) Clinically relevant fatigue is defined as FACIT-F $\leq$ 30 units. *Abbreviations*: D-12, Dyspnoea-12; MDP, Multidimensional Dyspnoea Profile; FACIT-F, Functional Assessment of Chronic Illness–Fatigue; CI, confidence interval.

**Table 3. Association for each breathlessness dimension with clinically relevant fatigue.**

|  | Association with fatigue, unadjusted | Association with fatigue, adjusted * |
|---|---|---|
|  | OR (95% CI) | OR (95% CI) |
| **D-12** |  |  |
| Total score | 3.02 (2.44 to 3.74) | 2.46 (1.79 to 3.38) |
| Physical score | 3.04 (2.45 to 3.77) | 2.56 (1.84 to 3.56) |
| Affective score | 2.37 (1.98 to 2.84) | 1.97 (1.50 to 2.58) |
| **MDP** |  |  |
| A1 unpleasantness score | 2.88 (2.30 to 3.60) | 1.85 (1.34 to 2.57) |
| Perception score | 2.69 (2.17 to 3.35) | 1.87 (1.37 to 2.56) |
| Emotional response score | 2.65 (2.17 to 3.25) | 1.61 (1.21 to 2.13) |

Associations are by logistic regression for each breathlessness score separately, adjusted for potential confounders (below). Clinically relevant fatigue is defined as FACIT-F $\leq$ 30 units.

\* Adjusted for the confounders smoking history, packet-years, BMI, respiratory diseases, cardiovascular diseases, depression, cancer, sleep apnoea and cardiac surgery.

*Abbreviations*: D-12, Dyspnea-12; MDP, Multidimensional Dyspnoea Profile; FACIT-F, Functional assessment of chronic illness–fatigue; BMI, body mass index; CI, confidence interval; OR, odds ratio.

## Strengths and limitations

Strengths of this study include the use of validated and established instruments for measuring multidimensional breathlessness and fatigue, a large sample size, and data on several relevant potential confounders. The main limitation is that our data only pertain to 73-year-old men, which may limit the generalizability of the findings. Given that women often report more breathlessness [39, 40], further studies are needed to see if there is an association between breathlessness dimensions and fatigue in women, and if there is a difference between the sexes in this respect. The participants were also relatively healthy, and few reported severe breathlessness–and only around 12% of the participants reported clinically relevant fatigue. Data on clinical evaluation of reversible underlying conditions for breathlessness or fatigue were not available, such as haemoglobin for anaemic status.

## Implications

Our findings have several implications. Both D-12 and MDP complement FACIT-F and provide useful information for our population–relatively healthy older men. We were not able to confirm our hypothesis that certain breathlessness dimensions are more strongly correlated to fatigue than others. When assessing a correlation with fatigue, both questionnaires seem to work equally well. MDP A1 only comprise a single descriptor compared to the 12 descriptors in D-12 (total), which may help avoid unnecessary burden of assessments in people with more severe illness. If the patients report a high value on dimensions correlated with anxiety or fear, then a treatment can be tailored for the patient. When a clinician identifies a patient with severe breathlessness, our data suggests that they should ask about fatigue and *vice versa*. Given the relationship that has been identified, practical plans can be put in place to try and address any fatigue. Simply identifying that this may not be a surprising association of symptoms can at times be reassuring.

Future studies should include both men and women and of different ages to further explore which of D-12 and MDP is more strongly associated with fatigue, and which of the breathlessness dimensions drives the association. It would also be important to explore the same associations in people living with more advanced conditions (who might experience more intense fatigue) and

those with more severe breathlessness. Another important topic for future studies is to see if breathlessness may be more associated with the degree of frailty than with fatigue [41].

## Conclusion

Breathlessness dimensions assessed using D-12 and MDP showed similar associations with a validated measure of fatigue in elderly men. Both breathlessness instruments are useful and can be used to evaluate associations with fatigue, with no specific breathlessness dimension score being more strongly correlated to fatigue in our population.

## Author Contributions

**Conceptualization:** Magnus Ekström.

**Data curation:** Max Olsson, Magnus Ekström.

**Formal analysis:** Lucas Cristea.

**Funding acquisition:** Magnus Ekström.

**Methodology:** Magnus Ekström.

**Supervision:** Magnus Ekström.

**Writing – original draft:** Lucas Cristea, Magnus Ekström.

**Writing – review & editing:** Lucas Cristea, Max Olsson, Jacob Sandberg, Slavica Kochovska, David Currow, Magnus Ekström.

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
