## [Decision Letter · Decision Letter 0]

21 Aug 2023

PONE-D-23-21052Which breathlessness dimensions associate most strongly with fatigue? - the population-based VASCOL study of elderly men.PLOS ONE

Dear Dr. Cristea,

Thank you for submitting your manuscript to PLOS ONE. After careful consideration, we feel that it has merit but does not fully meet PLOS ONE’s publication criteria as it currently stands. Therefore, we invite you to submit a revised version of the manuscript that addresses the points raised during the review process.

We look forward to receiving your revised manuscript.

Kind regards,

Ari Samaranayaka, PhD

Academic Editor

PLOS ONE

Journal Requirements:

2.Our internal editors have looked over your manuscript and determined that it is within the scope of our Aging in Human Health and Disease Call for Papers. This call for papers aims to highlight the excellent work being done by researchers across the world on the subject of aging. Additional information can be found on our announcement page: https://collections.plos.org/call-for-papers/aging-in-human-health-and-disease/. If accepted, your submission will be included within the collection. Please note that being considered for the Collection does not require an additional peer review beyond the journal’s standard process and will not delay the publication of your manuscript if it is accepted by PLOS ONE. If you have any questions or concerns about this process, please contact the journal at plosone@plos.org

"The study was funded by unrestricted grants from the Research Council of the Region of Blekinge, and from the Swedish Research Council (Dnr: 2019-02081).

Funders did not affect the design, conduct, analysis or reporting of the study in any way."

"Dr. Currow is an unpaid advisory board member for Helsinn Pharmaceuticals. He is a paid consultant and receives payment for intellectual property with Mayne Pharma and a consultant with Specialised Therapeutics Australia Pty. Ltd.

The remaining authors report no competing interests in any way."

Additional Editor Comments:

• The research aims need to be better justified by elaborating possible impact of the findings. Need to say why/how the assessment and treatment of patients can be influenced in future by when knowing different dimensions of breathlessness are associated with fatigue.

• Authors say “All relevant data are within the manuscript and its Supporting Information files”. I haven’t seen raw data; are you referring to software outputs in tables S1 and S2? If raw data cannot be made public, please say the reasons.

• Supplementary files. S1 liner regression and S2 logistic regression file simply presented Stata software outputs in a way not meaningful for readers who are unfamiliar with abbreviated variable names in analysis dataset and their categorisations. Reference groups also not disclosed. Please re-format it in a reader-friendly way.

• Brief description of the study design is helpful even if it is previously published.

• Table1. when calculating mean(sd) packyears, those nonsmoking currently were treated as zeros or N/A ? Should the respiratory disease total 65 changed to 57? Should the cardiovascular disease total 274 changed to 200?

• Table1. methods says MDP A1 unpleasantness score is in 0 to 10 scale. If so how can mean and SD larger than 10 for fatigue and non-fatigue groups in table1? Also, how can the mean for “all” is much smaller than the means in its component groups?

• Based in table4, authors say, “… only D-12 was independently associated with fatigue”. I assume table4 presented beta coefficients from OLS regression, and above conclusion is because CI for D12 being negative. Can you assess the independent associations of D12 and MDP by using them as predictors in the same model when they are 2 different measures for the same thing (breathlessness)? D12 and MDP are likely be highly correlated, then it is not surprising one of them to become non-significant when accounted for each other.

• Figure3 caption says clinically relevant fatigue level is Facit-F ≤30. Fatigue is log transformed and Z transformed before using them in linear regression. Then, how can we read having (or not having) of this clinically relevant fatigue level ?

Reviewers' comments:

Reviewer's Responses to Questions

**Comments to the Author**

1. Is the manuscript technically sound, and do the data support the conclusions?

Reviewer #1: Partly

Reviewer #2: Yes

2. Has the statistical analysis been performed appropriately and rigorously? 

Reviewer #1: Yes

Reviewer #2: Yes

3. Have the authors made all data underlying the findings in their manuscript fully available?

Reviewer #1: Yes

Reviewer #2: Yes

4. Is the manuscript presented in an intelligible fashion and written in standard English?

Reviewer #1: Yes

Reviewer #2: Yes

5. Review Comments to the Author

Reviewer #1: Research to improve understanding and awareness of the inter-relationship between breathlessness and fatigue is important. This has been highlighted by the Long Covid population where the 2 symptoms often co-exist and potentiate each other. However, although this study fulfils criteria for publication, I am not sure what it adds to the scientific and medical community. The main question relates to whether particular dimensions of subjective breathlessness questionnaires predict or are associated with fatigue. The limitations as I see are:

1) Why is it helpful to use a breathlessness questionnaire to understand fatigue? Would not asking about fatigue or using the FACTIT-F (or similar) be a better/more straightforward method?

2) Why chose a healthy cohort? Would it not make more sense to select a breathless group and study breathlessness questionnaires to see how they relate in this preselected population? Although 677 participants took part, only 79 had fatigue as a symptom so number relatively small in which to draw conclusions

3) The authors have stated that those with fatigue are more breathless – this seems intuitive given the increased effort put into work of breathing

4) The authors state that assessment and treatment of patients may be influenced by determining whether different dimensions of breathlessness are associated with fatigue. No example of such has been given and I cannot think of one myself. All clinical assessments of breathless patients should consider confounding factors that can increase perception of breathlessness – including pain, mood, co-morbidities and fatigue. I do not think delineating areas on a breathlessness questionnaire is the way to do this.

Reviewer #2: Authors described a cross-sectional study aimed the possible correlation of perceived fatigue with dyspnea, using different multiparametric scales in a population of elderly males. The aim seems interesting, and I consider the research area promising, although the work has several limitations that prevent the generalization of the results.

Minor revisions

Insert captions/footnotes for the images.

Insert in the supplementary materials all the scales evaluated and relative bibliographic references.

There is no mention in the text of the results shown in Figure 3.

Major revisions

1) Explain the choice of FACIT-F over other scales (e.g. VAS or ROF). Front Phys 2018;9:1285.

2) Explain the cut-off of 30 intended as a significant effort and the reason for this choice, possibly with an indication of the bibliographic reference.

3) In the selected population, only 11.7% demonstrated significant fatigue, as defined (i.e. FACIT-F <30) compared to 33% of dyspneic patients. Is it possible to hypothesize that the fatigue is not adequately described or that it is necessary to have a different grading?

4) Were there any cancer patients represented in the selected population? Was there a history of chemotherapy or any active or previous therapy with potentially myopathic drugs?

5) Are blood chemistry data available? such as hemoglobin? Anemic status could be an influential parameter, both on dyspnea and degree of fatigue, regardless of comorbidities. What Authors consider about this issue, this could be a point for discussion.

6) Are more data available about heart failure patients and their functional class? Were any of the patients represented in the study receiving oxygen therapy?

7) Authors wrote: “When looking at what each dimension adds to the model it is very similar and ranges between 54% and 60%”. Please, describe better this sentence.

8) Authors wrote: “Breathlessness was associated with increasing fatigue across all dimension scores in elderly men, but no breathlessness dimension was more strongly associated with fatigue than the others and the associations were only modest”. I believe the section of the discussion should be expanded, bringing more hypotheses for the interpretation of results and evaluating any further feasible insights. Is it possible to generalize in view of the type of study and the limited population selected?

9) Furthermore, how do Authors consider the choice of a population of males only to be possibly influential in terms of results? Is it reasonable to expect different results if female subjects were also included? In the literature, although there are no large population studies, the prevalence of fatigue and dyspnea is mostly reported in female subjects (Advances in Medical Sciences 2019; 64(2):303-308; Curr Opin Pulm Med 2017;23(2):117-122).

10) As a possible future research and insights, do Authors believe that dyspnea may be more associated with the degree of frailty than with fatigue alone in the population? (Int J Chron Obstruct Pulmon Dis 2020; 15: 1349–1356).

6. PLOS authors have the option to publish the peer review history of their article (what does this mean?). If published, this will include your full peer review and any attached files.

Reviewer #1: No

Reviewer #2: No

---

## [Author Response · Author response to Decision Letter 0]

10 Oct 2023

Editor comments:

1. The research aims need to be better justified by elaborating possible impact of the findings. Need to say why/how the assessment and treatment of patients can be influenced in future by when knowing different dimensions of breathlessness are associated with fatigue.

We agree with this suggestion, and we added the following texts in the Discussion at page 13: “If, for example, the patients report a high value on dimensions correlated with anxiety or fear then a treatment can be tailored for the patient”. 

“In clinical consultations, if a person identifies severe breathlessness, then these data suggest that the clinician should ask about fatigue and vice versa. Given the relationship that has been identified, practical plans can be put in place to try and address any fatigue. Simply identifying that this may not be a surprising association of symptoms can at times be reassuring”. 

2. Authors say “All relevant data are within the manuscript and its Supporting Information files”. I haven’t seen raw data; are you referring to software outputs in tables S1 and S2? If raw data cannot be made public, please say the reasons.

The VASCOL research group will consider requests for obtaining pseudonymized data from the VASCOL study by external collaborators upon reasonable request. All requests must also be approved by the Swedish National Ethical Review Board and are not allowed to be public available.

3 Supplementary files. S1 linear regression and S2 logistic regression file simply presented Stata software outputs in a way not meaningful for readers who are unfamiliar with abbreviated variable names in analysis dataset and their categorisations. Reference groups also not disclosed. Please re-format it in a reader-friendly way.

The regression output has been removed as supplementary files.

4 Brief description of the study design is helpful even if it is previously published.

The follow text has been added under Methods on page 4. “In 2010-2011, 1900 men aged 65 years in Blekinge, Sweden were invited to a screening campaign of abdominal aortic aneurysm – those men were invited to participate in the VASCOL study (ongoing longitudinal epidemiological cohort study). VASCOL was based on physiological measurements, but also completion of self-reported surveys – including information about multidimensional breathlessness and fatigue (through D-12, MDP and FACIT-F). Data was collected in 2010-2011, and the same men (who were still alive and with a known address) were asked in 2019 to respond to a follow-up postal survey”. 

5 Table1. When calculating mean(sd) packyears, those nonsmoking currently were treated as zeros or N/A ? Should the respiratory disease total 65 changed to 57? Should the cardiovascular disease total 274 changed to 200?

We thank the reviewer for drawing attention to this: non-smokers were treated as zeros, and Table 1 has been updated to show the correct numbers for respiratory and cardiovascular diseases. 

6 Table1. Methods says MDP A1 unpleasantness score is in 0 to 10 scale. If so how can mean and SD larger than 10 for fatigue and non-fatigue groups in table1? Also, how can the mean for “all” is much smaller than the means in its component groups?

It is much appreciated that this mistake was detected, and we thank the editor for this – the analysis has been performed again and table 1 updated with the correct values.

7 Based in table4, authors say, “… only D-12 was independently associated with fatigue”. I assume table4 presented beta coefficients from OLS regression, and above conclusion is because CI for D12 being negative. Can you assess the independent associations of D12 and MDP by using them as predictors in the same model when they are 2 different measures for the same thing (breathlessness)? D12 and MDP are likely be highly correlated, then it is not surprising one of them to become non-significant when accounted for each other.

We agree with the Editor's evaluation. This table has been removed since we cannot assess the independent associations due to multicollinearity.

8 Figure3 caption says clinically relevant fatigue level is Facit-F ≤30. Fatigue is log transformed and Z transformed before using them in linear regression. Then, how can we read having (or not having) of this clinically relevant fatigue level ?

The fatigue score was not transformed, and only the breathlessness scores were log- and Z-transformed. We have now clarified this in the Method section at page 6.

Reviewer 1 comments:

1) Why is it helpful to use a breathlessness questionnaire to understand fatigue? Would not asking about fatigue or using the FACTIT-F (or similar) be a better/more straightforward method?

We wanted to see if there is an association between breathlessness and fatigue – we have showed that there is an association, doctors can be aware of this and ask their breathless patients about their fatigue. Breathlessness consists of multiple dimension scores, which can be measured by D-12 and MDP and some symptom dimensions may be more strongly linked than others to worse fatigue. No study to date has evaluated this relationship.

2) Why chose a healthy cohort? Would it not make more sense to select a breathless group and study breathlessness questionnaires to see how they relate in this preselected population? Although 677 participants took part, only 79 had fatigue as a symptom so number relatively small in which to draw conclusions.

The study design is population based cross-sectional, which means that we include both healthy and ill participants. This enables us to draw conclusions that can be generalised for other similar populations. For example, we can examine strength of association between breathlessness and fatigue in the population and address relevant confounders such as health conditions prevalent in the older male population.

3) The authors have stated that those with fatigue are more breathless – this seems intuitive given the increased effort put into work of breathing.

Although this may seem intuitive, it is important to quantify the strength of this relationship. The clinical relevance is that anytime that one of these symptoms is identified in history taking, the clinician should enquire about the other symptom. Both symptoms are largely ignored, often due to a therapeutic nihilism on the part of clinicians, concerned that there may be little that they can do to address the symptoms. On the contrary, the evolving evidence base of both symptoms suggest that practical measures can be taken to reduce the impact of these symptoms. 

4) The authors state that assessment and treatment of patients may be influenced by determining whether different dimensions of breathlessness are associated with fatigue. No example of such has been given and I cannot think of one myself. All clinical assessments of breathless patients should consider confounding factors that can increase perception of breathlessness – including pain, mood, co-morbidities and fatigue. I do not think delineating areas on a breathlessness questionnaire is the way to do this.

If people score highly on the affective domain of breathlessness, for example depression, an appropriate treatment can be initiated such as cognitive behavioural therapy or antidepressants – a combination of both is probably the best treatment. 

Future research should seek to understand whether the affective component of breathlessness (unpleasantness) is more closely associated with fatigue, whether it is the intensity (severity) or both. This may help to delineate phenotypes at great risk of fatigue and potentially alert clinicians to have a lower threshold for inquiring about fatigue.

Reviewer 2 comments:

1. Explain the choice of FACIT-F over other scales (e.g. VAS or ROF). Front Phys 2018;9:1285.

FACIT-F is a validated tool that is suited to population studies such as this and it enables us to compare our findings with other studies. Although each tool has its strengths and weaknesses, FACIT-F is a psychometrically robust tool that was a reasonable tool to include in the questionnaire in the VASCOL study. It also measures multiple dimensions of fatigue in comparison to, for instance, a VAS. 

2. Explain the cut-off of 30 intended as a significant effort and the reason for this choice, possibly with an indication of the bibliographic reference.

The cut-off of 30 was chosen as significant because it was more than one standard deviation lower than the average fatigue score in the general population, thus indicating a statistically and clinically significant score (Piper, B.F. and D. Cella, Cancer-Related Fatigue: Definitions and Clinical Subtypes. Journal of the National Comprehensive Cancer Network J Natl Compr Canc Netw, 2010. 8(8): p. 958-966.)

Norman et al have also shown that even one half of standard deviation being clinically significant when the threshold for clinical significance is unknown (Norman GR, Sloan JA, Wyrwich KW. Interpretation of changes in health-related quality of life: the remarkable universality of half a standard deviation. Med Care. 2003 May;41(5):582-92. doi: 10.1097/01.MLR.0000062554.74615.4C. PMID: 12719681.)

3) In the selected population, only 11.7% demonstrated significant fatigue, as defined (i.e. FACIT-F <30) compared to 33% of dyspneic patients. Is it possible to hypothesize that the fatigue is not adequately described or that it is necessary to have a different grading?

A higher cut off score would yield more patients with clinically significant fatigue; it was shown in a study of anaemic cancer patients that a cut off score of 43 or less best distinguished the anaemic cancer patients from those in the general population. FACIT-F is a validated tool that is suited to population studies such as this and it enables us to compare our findings with other studies. It is a psychometrically robust tool that was a reasonable tool to include in the questionnaire in the VASCOL study.

4) Were there any cancer patients represented in the selected population? Was there a history of chemotherapy or any active or previous therapy with potentially myopathic drugs?

There were 109 patients with a history of cancer in this population sample, but more detailed data on the condition(s) and treatment are not available. 

5) Are blood chemistry data available? such as hemoglobin? Anemic status could be an influential parameter, both on dyspnea and degree of fatigue, regardless of comorbidities. What Authors consider about this issue, this could be a point for discussion.

Unfortunately, we did not have any data regarding blood chemistry and haemoglobin available. Anaemia can influence both breathlessness and fatigue and it is a common occurrence in patients, in both primary care and in a hospital setting. This has been clarified in the strength and limitations section of the Discussion on page 13.

6). Are more data available about heart failure patients and their functional class? Were any of the patients represented in the study receiving oxygen therapy?

We do not have any more data about the heart failure patients and their functional classification. Only 9 patients in our sample received oxygen therapy.

7). Authors wrote: “When looking at what each dimension adds to the model it is very similar and ranges between 54% and 60%”. Please, describe better this sentence.

For the regression models with D-12 and fatigue, 59-60% of the variance in fatigue scores was explained by the models. For the regression models with MDP and fatigue, 54-56% of the variance was explained by the models - the remaining percentage is due to the factors we did not measure. The variance measured in the models were adjusted for confounders – this has further been clarified in both Methods on page 7 and in the captions for table 2 on page 10.

 

8). Authors wrote: “Breathlessness was associated with increasing fatigue across all dimension scores in elderly men, but no breathlessness dimension was more strongly associated with fatigue than the others and the associations were only modest”. I believe the section of the discussion should be expanded, bringing more hypotheses for the interpretation of results and evaluating any further feasible insights. Is it possible to generalize in view of the type of study and the limited population selected?

We have quite a large sample size and a lot of data which allows us to address several relevant confounders – our findings give some indication about the relationship between breathlessness and fatigue. It is quite difficult to generalize our findings since our population was relatively healthy and consisted only of 73-year-old men which has been mentioned as a limitation under the Discussion section page 12. 

9) Furthermore, how do Authors consider the choice of a population of males only to be possibly influential in terms of results? Is it reasonable to expect different results if female subjects were also included? In the literature, although there are no large population studies, the prevalence of fatigue and dyspnea is mostly reported in female subjects (Advances in Medical Sciences 2019; 64(2):303-308; Curr Opin Pulm Med 2017;23(2):117-122).

It is an issue that only men were part of the study and if women also were involved it is possible that we would have seen different findings to some degree. However, men and women have generally been found to have similar associations between factors (self-reported outcomes and physiological responses) in studies, including a similar relation for key physiologic factors such as the ventilatory drive, level of ventilation and exertional breathlessness.( Ekström M, Sundh J, Schiöler L, Lindberg E, Rosengren A, Bergström G, Angerås O, Hedner J, Brandberg J, Bake B, Torén K. Absolute lung size and the sex difference in breathlessness in the general population. PLoS One. 2018 Jan 5;13(1):e0190876. doi: 10.1371/journal.pone.0190876. PMID: 29304074; PMCID: PMC5755925.) 

Therefore, we would expect quite similar associations between breathlessness and fatigue among women as well, but we agree with the reviewer that this is an open question, and this is highlighted in the Discussion as an area of future research.

10) As a possible future research and insights, do Authors believe that dyspnea may be more associated with the degree of frailty than with fatigue alone in the population? (Int J Chron Obstruct Pulmon Dis 2020; 15: 1349–1356).

The reference study about frailty and COPD is an interesting and important study and has been added as a reference and mentioned as an area for future research in the Discussion section page 13. 

Buarque GLA et al (Buarque GLA, Borim FSA, Neri AL, Yassuda MS, Melo RC. Relationships between self-reported dyspnea, health conditions and frailty among Brazilian community-dwelling older adults: a cross-sectional study. Sao Paulo Med J. 2022 May-Jun;140(3):356-365. doi: 10.1590/1516-3180.2021.0237.R2.27072021. PMID: 35508002; PMCID: PMC9671253.) showed an association between self-reported dyspnoea and frailty in community-dwelling older adults. 

It is possible that dyspnoea is more associated with frailty than fatigue in the population since dyspnoea affects 25-32% of the elderly (>70 years of age) in their daily life. (Smith, A.K., et al., Prevalence and Outcomes of Breathlessness in Older Adults: A National Population Study. J Am Geriatr Soc, 2016. 64(10): p. 2035-2041.)

---

## [Decision Letter · Decision Letter 1]

8 Nov 2023

PONE-D-23-21052R1Which breathlessness dimensions associate most strongly with fatigue? - the population-based VASCOL study of elderly men.PLOS ONE

Dear Dr. Cristea,

Thank you for submitting your manuscript to PLOS ONE. After careful consideration, we feel that it has merit but does not fully meet PLOS ONE’s publication criteria as it currently stands. Therefore, we invite you to submit a revised version of the manuscript that addresses the points raised during the review process.

We look forward to receiving your revised manuscript.

Kind regards,

Ari Samaranayaka, PhD

Academic Editor

PLOS ONE

Journal Requirements:

Additional Editor Comments:

• In the previous review I pointed out some inconsistency in numbers in table1. Authors responded saying it was updated to show the correct numbers. However I can still see some inconsistency. For example, cardiovascular disease total is not the same as the sum of those with fatigue and non-fatigue.

• In the previous review I queried on the Figure3 caption that refers to identifying clinically relevant fatigue level of Facit-F ≤30 using beta coefficients when both X and Y data are transformed twice before using in OLS regression. Authors responded adding a sentence to the statistical analysis section saying only the X data were transformed, Y data were not transformed. However I can still see the previous statement in the same paragraph that says both breathlessness and fatigue were transferred. Also, figure3 caption also says all scores were analysed as Z scores. the same. Authors need to proof read the manuscript to avoid contradictory statements.

Reviewers' comments:

Reviewer's Responses to Questions

**Comments to the Author**

1. If the authors have adequately addressed your comments raised in a previous round of review and you feel that this manuscript is now acceptable for publication, you may indicate that here to bypass the “Comments to the Author” section, enter your conflict of interest statement in the “Confidential to Editor” section, and submit your "Accept" recommendation.

Reviewer #2: All comments have been addressed

2. Is the manuscript technically sound, and do the data support the conclusions?

Reviewer #2: Yes

3. Has the statistical analysis been performed appropriately and rigorously? 

Reviewer #2: Yes

4. Have the authors made all data underlying the findings in their manuscript fully available?

Reviewer #2: Yes

5. Is the manuscript presented in an intelligible fashion and written in standard English?

Reviewer #2: Yes

6. Review Comments to the Author

Reviewer #2: I am satisfied with the answers provided by the Authors and how the paper has been implemented following all the comments and requests of explanation suggested by Editor and Reviewers.

7. PLOS authors have the option to publish the peer review history of their article (what does this mean?). If published, this will include your full peer review and any attached files.

Reviewer #2: No

---

## [Author Response · Author response to Decision Letter 1]

3 Dec 2023

The reference list has been reviewed and all the references are correct - none of the 

references have been retracted.

2. In the previous review I pointed out some inconsistency in numbers in table1. Authors responded saying it was updated to show the correct numbers. However I can still see some inconsistency. For example, cardiovascular disease total is not the same as the sum of those with fatigue and non-fatigue.

We have gone through the numbers in table1 and have updated it for the correct numbers.

3. In the previous review I queried on the Figure3 caption that refers to identifying clinically relevant fatigue level of Facit-F ≤30 using beta coefficients when both X and Y data are transformed twice before using in OLS regression. Authors responded adding a sentence to the statistical analysis section saying only the X data were transformed, Y data were not transformed. However I can still see the previous statement in the same paragraph that says both breathlessness and fatigue were transferred. Also, figure3 caption also says all scores were analysed as Z scores. the same. Authors need to proof read the manuscript to avoid contradictory statements.

This has been addressed in the statistical analysis section and in the figure 3 captions – we greatly appreciate and thank the editor for all the comments.

---

## [Editor Report · Decision Letter 2]

5 Dec 2023

Which breathlessness dimensions associate most strongly with fatigue? - the population-based VASCOL study of elderly men.

PONE-D-23-21052R2

Dear Dr. Cristea,

We’re pleased to inform you that your manuscript has been judged scientifically suitable for publication and will be formally accepted for publication once it meets all outstanding technical requirements.

Kind regards,

Ari Samaranayaka, PhD

Academic Editor

PLOS ONE
---

## [Editor Report · Acceptance letter]

11 Dec 2023

PONE-D-23-21052R2 

Which breathlessness dimensions associate most strongly with fatigue? – the population-based VASCOL study of elderly men 

Dear Dr. Cristea:

I'm pleased to inform you that your manuscript has been deemed suitable for publication in PLOS ONE. Congratulations! Your manuscript is now with our production department. 

Kind regards, 

on behalf of

Dr. Ari Samaranayaka 

Academic Editor

PLOS ONE